# Analysis of the Generation of Vibration Signals under Uniaxial Loading of Materials Using the Coherent Properties of Laser Radiation

**DOI:** 10.3390/ma13092046

**Published:** 2020-04-27

**Authors:** Petr Louda, Artem Sharko

**Affiliations:** 1Department of Material Science, Technical University of Liberec, 461 17 Liberec, Czech Republic; 2Department of Automation and Computer-Integrated Technologies, Kherson National Technical University, 73008 Kherson, Ukraine

**Keywords:** laser vibrometry, tensile diagram, information processing, discrimination levels, Fourier transform

## Abstract

The present article describes the results of theoretical and experimental investigations of the force response of materials to external tensile stresses. The method used is based on remote precision measurements of the amplitudes of the harmonics of vibration signals and synchronous fixation of deformations under load. It was determined that the shape of the propagating acoustic signal depends not only on the bias time, but also on the frequency. In addition, fixation of the moments of occurrence of the vibrational signals and determination of the number of excesses in the amplitudes of harmonics over the discrimination level allows structural models to be studied in order to diagnose the strength properties of materials under dynamic loading of solids. The experimental setup consisted of a P100 Labtest-2 tearing machine providing a loading speed of 6.22 mm/min, a Polytech PSV–400 vibrometer including lasers, and a computer. Measurements were made at frequencies of 0.4, 1.6, and 40 kHz. An analysis of the mathematical models of the occurrence and propagation of acoustic signals in a material under load is presented, and the features of their application are reported. Transferring the moments of occurrence of vibrational signals to a strain diagram from the applied load allows the development of physical processes of hardening and destruction of materials to be traced. The occurrence of vibrational signals under load can be used as an information parameter for the diagnosis of developing defects in the structure of materials under load.

## 1. Introduction

The study of the kinetics of the deformation of materials is associated with the need to analyse the force response of the material on external tensile stresses [1,2]. Because of the complexity of the relationships of structural parameters and the physical and mechanical characteristics of materials, one cannot rely on a priori information about these changes [3,4]. For this, it is necessary to conduct an experimental investigation of the deformation of materials during uniaxial loading with simultaneous recording of signal emergence moments in real time [5].

Acoustic methods hold a special place among methods for assessing the state of metal structures of various shapes and sizes, as they apply the use of special diagnostic procedures to assess operational reliability without disassembling the structure itself. Much attention is paid to the diagnosis of damage and monitoring the development of defects. In [6] a Kalman filter-based algorithm was used to measure the parameters of moving objects; in [7] diagnostic features were used for the condition monitoring of hypoid gear utilising the wavelet transform; in [8] location acoustic signals were used in a power transformer using fuzzy adaptive particle swarm optimisation; in [9] improved least square generative adversarial networks were applied for rail crack detection using the acoustic emission technique; in [10] energy distribution and fractal characterisation of acoustic emissions during coal deformation and fracturing were employed. Acoustic methods are based on the registration of stress waves arising as a result of loading and destruction of the structure of materials [11,12].

Improving the technical diagnostics allows not only defects to be detected and the degree of their danger to established, but also the stress-strain state of the material to be evaluated [13,14]. Under these conditions, the use of the vibroacoustic method is promising, since the diagnostic signals contain information about the dynamics of the processes [15].

The vibration diagnostics method allows the degree of malfunction of controlled objects to be determined by the nature of the vibration. The applicability of the method for solving specific tasks has been reviewed in a number of studies: in [16] rolling element bearing defect frequencies were verified by vibration measurements on bearings with artificial faults on the outer/inner rings; in [17] vibration diagnostics methods and programs were applied to assess the remaining lifetime of oil equipment; in [18] vibration-based diagnostics of epicyclic gearboxes and soft-computing methods were employed; in [19] a system of compressor units defects were diagnosed by the vibration value; in [20] vibration diagnostics on a hydraulic rotator were described; in [21] a gas turbine was introduced and vibration diagnostic basics; and in [22] vibration diagnostics of plates of knife refiners were performed. However, the complexity of the trajectories of the change nature in the vibrational signal does not allow this method to be used for quantitative assessment of the stress state of a material.

Remote precision measurements of moving materials using the coherent properties of laser radiation make it possible to perform static measurements of the relative movements of structural elements before and after loading. Various technical improvements and original solutions are offered. In [23] optical coherence tomography was carried out with real-time profilometry of multiple inner layers; in [24] monitoring of steel shaft headgears were performed; in [25] symmetrised dot pattern and density-based clustering was carried out; in [26] frequency ratios to diagnose structural damage were used; in [27] auto-encoders were used; in [28] deep random forest fusion was used; in [29] polymer composite materials were developed; and in [30] the full spectrum and a dual channel was measured. However, there is no information on dynamic measurements of the stress-strain state of materials in real time.

The disadvantages of using existing methods of vibration diagnostics are the complex data processing procedure, the periodic mode of measuring the state of defects, which requires the loading device to be stopped, and the stationary nature of the loading during testing.

The idea behind the present work was to link together three important scientific and practical results:vibration diagnostics,fixation of the movements with the help of laser vibrometers with computer registration of signal processing,interrelation of acoustic signals with characteristic sections of the loading diagram.

The objective of the study was to create a method for measuring the stress-strain state of a material. Due to the technological features of this method, it would be possible to provide a continuous observation mode, simplify the data processing procedure, and increase the information content of the measurements.

The main aim of this work was to develop a non-contact method for determining the moments of vibration occurrence in different ranges of material loading using laser fixation in order to determine the technical state of the object. 

## 2. Materials and Methods

Theoretical studies of the occurrence of acoustic signals with changes in the structure of materials are developed in two main areas: the development of discrete models and models of the elastic continuum. Continuous models reflect free damped vibrations when scattered over local inhomogeneities and the propagation boundaries of acoustic signals. The periodicity of the crystal lattice, its structure, and the constitution of atoms determine the features of the energy spectra. With a known force field, it is possible to obtain information on changes in the structure of materials, which cannot be directly measured, by solving the equations of motion.

In this work, we used the method of coherent properties of laser studies, which allows remote precision measurements to be taken of deformations and vibrations of a solid using a laser Doppler vibrometer by expanding the vibration signals in a spectral series using direct Fourier transform and measuring the harmonic amplitudes. The measurements are taken in a continuous dynamic mode over the entire range of the load diagram, the moments of occurrence of vibrational signals are recorded, and the number of excesses of the harmonics amplitudes over the level of discrimination in the parts of the load diagram is determined.

The shape of the propagating acoustic signal depends not only on the bias time but also on the frequency *ω*.

The method is applied using the following developed device (Figure 1).

The output laser beam (a) is divided by a translucent mirror (5) into two equal-intensity beams (b and c), one of which (b) is directed through a translucent mirror (6) and lens (7) towards the test sample (8), placed in a loading device (3). The beam (d) is reflected from the surface of the test sample using the lens (7) and the translucent mirror (6) and is directed to the translucent mirror (10), where it is connected to the beam (c), which reaches here after being reflected from the translucent mirrors (5)(9). The total beam of interfering rays (cd) falls on the measuring photodetector (11).

During the deformation of the sample, the light flux and the beam (d) reflected from it change the frequency due to the Doppler effect. The interference of the light beam (c) reflected from a fixed mirror and the light beam (d) that has undergone a Doppler shift leads to the appearance of low-frequency beats in the intensity of the light flux (cd), which are converted into an electrical signal in the photodetector (11). This signal is transmitted to a personal computer equipped with software for receiving and processing the signal. The registered numerical signal is generated in the form of a file, which is recorded in the computer’s memory. The file is generated every time a vibration signal occurs.

The studied materials used were sheet steel, basalt fibre, dry wood, and plastic.

The analysis of the generation of vibration signals under the uniaxial loading of structural materials is based on Fourier transforms.

The experimental setup consisted of a P100 Labtest-2 tearing machine providing a loading speed of 6.22 mm/min, a Polytech PSV–400 vibrometer including lasers, and a computer. An image of the experimental setup is shown in Figure 2.

During the experiment, the laser beam recorded a change in the vibration of the sample in the horizontal plane, as shown in Figure 3.

Fixation of the moments of occurrence of the vibration signals in real time without stopping the bursting machine was performed by means of a computer, to which the signals were fed from the processor (Figure 4).

After the analogue-to-digital conversion, the electrical signals were fed to the computer.

The information-measuring system used in the experiment provides indication, recording and pre-processing of the signal with its further preservation in the computer’s memory for the subsequent post-processing of the received data and their visualisation in real time.

## 3. Results

The quantitative values of the occurrence of moments of the vibration signals in the coordinates of the absolute elongation of the samples with their reflection on the tension diagram for sheet steel are presented in Figure 5 and Table 1. The sample number is marked horizontally, and the measurement number is marked along the vertical line. Measurements were made at frequencies of 0.4, 1.6, and 40 kHz.

The occurrence moments of the vibration signals were grouped in accordance with the reference data for mild steels. Digits: I—the plot of the elastic region, II—the area of the plastic flow of the material, III—the phase of hardening, IV—the phase, prior to the rupture of the sample.

Despite the difference in the quantitative values of the breaking loads of the test samples, the general tendency towards an increase in strain with increasing stress is clearly visible.

The shape of the vibration signals in the areas of strain hardening of the sheet steel sample is shown in Figure 6, Figure 7, Figure 8 and Figure 9.

The signals obtained are characterised by a different number of oscillations that correlate with changes in the structure of the material associated with the discrete displacement of dislocations. The transfer of the moments of the appearance of vibration signals to the deformation rate diagram from the applied load makes it possible to trace the development of the physical processes of the hardening and destruction of the material. The primary location of vibration signals begins in the region of elastic deformation, and then, at the transition stage from the elastic to the plastic region, the frequency of the appearance of vibration signals increases, acquiring its maximum value in the region of the preceding destruction.

The results of the measurement of quantitative values of absolute elongation corresponding to the moment of occurrence of the vibration signals used for other materials also confirmed this effect (see Table 2).

In the nature of the presented time dependences of the vibrational signals, it is possible to distinguish a high-frequency component associated with oscillations inside the signal and a low-frequency component characterising the envelope of the signal. The signal envelope contains local extremes. The area under the envelope of the signal may be correlated with the energy of the process. Peak amplitudes correspond to the absolute maximum of the signal.

For a more comprehensive study of the observed effect, the form of the vibration signals was studied in the sections of the stretching diagram.

An effective way of analysing continuous acoustic signals is the Fourier transform, in which the signal decomposes into a basis of sines and cosines of different frequencies. The conversion coefficients are found by computing the scalar product of the signal with complex exponentials.
(1)$$F(\Omega )=\int_{-\infty }^{+\infty }f(t)e^{-j\Omega t}dt,$$
where *f*(*t*) is the signal and *F*(Ω) is the Fourier transform.

According to the uncertainty principle, the more the function is concentrated in time, the more it is blurred in the frequency domain [31]. When the function scale changes, the product of the probability density of the time and frequency ranges remains constant. Therefore, the measurement results are presented in the form of time (envelope of the signal) and frequency (Fourier transform of the signal) dependences of the signals under load.

The results of the Fourier transform are shown in Figure 10, Figure 11, Figure 12 and Figure 13.

The density of the frequency signal was determined by counting the number of intersections of the threshold level of the signal divided by the frequency range of the signal. The threshold level was considered equal to 0.2*C_s_*_max_ for each signal, where *C_s_*_max_ is the maximum amplitude of the Fourier transform. The choice of the threshold signal level within 20% of the maximum amplitudes during the measurement of the time and frequency characteristics is due to significant noise of the signals. 

Analysis of the distribution of the amplitudes of the vibrational signals while increasing the applied load made it possible to isolate the necessary level of discrimination. An informative parameter of the vibration signal is the total number of pulses defined as the number of pulse excesses above the discrimination level during the observation period. When processing the experimental data, the Mathematica 9.0 computer mathematics system and algorithms for working with arrays of numerical data were used to find the maximum (minimum) elements of the array, sort the data of the array by characteristic, combine the data, and spline the interpolation.

The distribution of the amplitudes of the vibration signals while increasing the applied load is shown in Figure 14.

The subsequent calculation of the number of excess points of *N* above the discrimination levels is shown in Figure 15.

Visualisation of the number of excesses of vibration signals over the discrimination levels, constructed from the data in Figure 15, reveals the maximum value that falls on the site preceding the gap (Figure 16).

This is in complete accordance with the changes in the frequency of the occurrence of vibration signals presented in Figure 5.

All of the presented characteristics of the vibrational signals and the Fourier transforms tend to increase with increasing strain. It is possible to unequivocally ascertain the increase in the energy of vibrational signals as they approach the zone of irreversible changes and destruction (zone III), which serves as an indicator of the pre-critical state of the test sample. With an increase in the applied load and an increase in the relative deformation of the samples, the formation of vibrational signals that correlate with deformation jumps occurs. Since the square of the amplitude of the acoustic signal is proportional to its power, with increasing load on the material and increasing the degree of relative deformation, the power of the acoustic signal increases.

## 4. Discussion 

A feature of the energy spectrum of the acoustic signals is the discrete nature of the structural changes and the continuous propagation of acoustic waves. All this makes the development of mathematical models of microstructural media relevant.

Study of the energy spectrum of signals of nanosized objects is explained by the importance of solving questions about the propagation of vibration characteristics preceding the destruction of materials. In recent years, new models of continuum mechanics have been intensively developed.

Theoretical studies of the occurrence of acoustic signals with changes in the material structure have been developed in two main areas: the development of discrete models and models of the elastic continuum.

Media exist with simple and complex structures. In environments of simple structures, the only kinematic variable is the displacement vector, which completely determines the state of the environment. To describe a medium with a complex structure, a set of motions and deformations of different orders is introduced, which characterise the internal degrees of freedom and force constants. The difference between media with simple and complex structures is manifested at relatively high frequencies, of the order of the natural frequencies of the internal degrees of freedom.

In [32], a one-dimensional model of a discrete microstructure is presented in the form of an unbounded linear chain of point masses connected by elastic bonds (Figure 17).

The force constants that denote the properties of such a discrete model are determined through the parameters of the elastic bonds. The effective characteristics of the bonds are found from the interaction potential of atoms [33].

The potential energy of such a chain, Φ, is a function of the displacement field *u*(*n*).
(2)$$\Phi =\Phi_{0}+\sum_{n}\Phi (n)u(n)+\frac{1}{2}\sum_{n,n\prime }\Phi (n,n\prime )u(n)u(n\prime )+\frac{1}{3!}\sum_{n,n\prime ,n\prime \prime }\Phi (n,n\prime ,n\prime \prime )u(n)u(n\prime )u(n\prime \prime )+\ldots ,$$
where Φ_0_ is the linear chain energy in equilibrium, and *n*, *n*’, and *n*’’ are the numbers of interacting particles.

The Lagrange function in this model is the law of conservation of energy.
(3)$$L=\frac{m}{2}\sum_{n}{\left[\frac{\partial u(n,t)}{\partial t}\right]}^{2}-\frac{1}{2}\sum_{n,n\prime }\Phi (n,n\prime )u(n,t)u(n\prime ,t)+\;\sum_{n}q(n,t)u(n,t),$$
where Φ(*n*,*n’*) is the force constants, and *q*(*n*,*t*) is the external forces.

When considering Equation (3), the equation of the vibrational motion of particles in a linear chain takes the form:(4)$$m\frac{\partial^{2}u(n,t)}{\partial t^{2}}+\sum_{n\prime }\Phi (n,n\prime )u(n\prime ,t)=q(n,t).$$

The physical meaning of the force constants Φ(*n*,*n’*) is visible from the equation of motion of the particles. If a particle with a coordinate *n’* receives a unit displacement, then an external force compensating for the elastic bond reaction, *Ψ*(*t*), is a characteristic of the interaction between atoms.

When defects occur, the energy required for the formation of dislocations with a length of one interatomic distance is equal to the energy required for the formation of one vacant site in the lattice. In the absence of other dislocations, it will move. Therefore, a transition is necessary from considering the vibrations of atoms in a discrete structure to a set of propagating waves (Figure 18). 

In a solid, the equilibrium positions of atoms are fixed. For wavelengths comparable to the interatomic distance, the medium in which the wave propagates cannot be considered continuous. Therefore, there is a frequency limit, *ω*_0_, defining the boundary of discrete transformations of the structure and continuous propagation of acoustic signals.

The energy spectrum of acoustic signals of a simple structure can be represented using Fourier transform.

In the Fourier transform, the acoustic signal is decomposed on the basis of sines and cosines of different frequencies. The shape of the acoustic signal depends not only on the bias time, but also on the frequency; therefore, along with the bias functions, their Fourier images should be considered:(5)$$u(\omega )=\int u(t)e^{i\omega t}dt,\;u(t)=\frac{1}{2\pi }\int u(\omega )e^{-i\omega t}d\omega .$$

The discrete structure is characterised by the short-range interaction of the constituent elements, while the continuous medium is characterised by the long-range interaction [34].

One of the most important concepts of acoustic signal propagation is the concept of a quasicontinuum, which allows discrete and continuous models to be considered (Figure 19).

The task of considering the one-dimensional continuum is to interpolate the displacement functions at the nodes and to choose from the whole set of interpolating functions, *u*(*x*) *+ Ψ*(*x*), the smoothest one to filter out the rapidly oscillating components of acoustic signals [35].

The analytical justification for the interaction of discrete factors initiating the appearance of acoustic signals with their continuous distribution in the medium is that each operation on the functions of a discrete argument can be associated with operations on their images—functions of a continuous argument. This will translate a continuous acoustic signal into a sequence of numbers [36].

The interpolation function will have the form:(6)$$u(x)=\frac{1}{2\pi }\int e^{ikx}u(k)dk.$$

In the Fourier transform, the argument changes and instead of *x* will be *k* and instead of *u*(*x*) will be *u*(*k*).

For function *u*(*x*), the Fourier transform will have the form:(7)$$u(k)=\int e^{-ikx}u(x)dx,$$
where *u*(*x*) and *u*(*k*) are considered as representations of the same function with different functional bases. 

The spectrum of acoustic signals is expressed through the Fourier transform, concentrated on a segment that determines the geometric dimensions of the unit cell of the microstructure under study.

The considered one-dimensional discrete-continuous model of the microstructure of a continuous medium in the form of an unlimited linear chain of point masses connected by elastic bonds allows us to obtain equations of motion and determine the force constants of the material. The model can be used if the deformation propagates rather slowly, on the scale of the interaction radius and cell size, so that the movement occurs not only in coordinates, but also in time.

Modern mathematical models of acoustic signals take into account the presence of several scales or structural levels in the medium, their coordinated interaction and the possibility of energy transfer from one level to another [37].

The use of the classical model of a continuous medium for studying energetic acoustic vibrational signals in microstructural environments encounters an uncertainty associated with the existence of internal motions of the structural components of the medium. Oscillations of atoms in the crystal lattice do not initiate a propagating wave. For its occurrence, external disturbances are necessary.

A discrete-continuous model of the energy spectrum of acoustic signals in complex structures with internal degrees of freedom is proposed, in which the translational and rotational invariance of energy in the energy spectrum of acoustic signals is established, which is also acceptable for vibration diagnostics.

Modelling the energy spectrum of a complex structure involves the allocation of a minimum volume in an array—a structural cell that displays the main features of the behaviour of the material [38] (Figure 20).

The Lagrange function in a diatomic cell is expressed by the equation:(8)$$L=\frac{1}{2}\sum_{n,j}m_{j}{\left[\frac{\partial w(n,j)}{\partial t}\right]}^{2}-\frac{1}{2}\sum_{n,n\prime ,j,j\prime }w(n,j)\Phi (n-n\prime ,j,j\prime )w(n\prime ,j\prime )+\sum_{n,j}f(n,j)w(n,j).$$
where *n* is the cell number, *m_j_*(*j* = 1,2) is the mass of particles in the cell, *w*(*n*,*j*) is the displacement of particles *j* in the cell *n*, Φ(*n* – *n*’,*j*,*j*’) is the force constant, and *f*(*n*,*j*) is the external force that determines the occurrence of the vibrational signals.

The physical meaning of the model becomes clear if we move on to new variables, such as:displacement of the centre of mass of the cell:
(9)$$u(n)=\frac{1}{m}\left[m_{1}\omega (n,1)+m_{2}\omega (n,2)\right],$$relative displacement of particles in the cell:
(10)$$\eta (n)=\frac{m_{1}\xi_{1}\omega (n,1)+m_{2}\xi_{2}\omega (n,2)}{I},$$
where *m*_1_ and *m*_2_ are the masses of atoms in the cell, *I* is the moment of inertia of the cell, and *ξ*_1_ and *ξ*_2_ are the coordinates of the particles in the cell relative to the coordinate of the centre of mass.

We obtain the equations of motion of the entire cell in a medium initiating the propagation of an acoustic signal:(11)$$m\frac{\partial^{2}u(n,t)}{\partial t^{2}}+\sum_{n\prime }\Phi^{00}(n-n\prime )u(n\prime )+\sum_{n\prime }\Phi^{01}(n-n\prime )\eta (n\prime )=q(n),$$
(12)$$I\frac{\partial^{2}\eta (n,t)}{\partial t^{2}}+\sum_{n\prime }\Phi^{10}(n-n\prime )u(n\prime )+\sum_{n\prime }\Phi^{11}(n-n\prime )\eta (n\prime )=\mu (n),$$
where *q*(*n*) and *μ*(*n*) are generalised forces corresponding to *u*(*n*) and *η*(*n*).

Matrix Φ*ss*’(*n*)(*s*,*s*’ = 0, 1) is expressed coordinate-wise through force constant Φ(*n*,*j*,*j*’) precursors of the AE signal [39]:(13)$$\Phi_{\left(n\right)}^{SS\prime }=\left(\begin{array}{cc} \Phi^{00} & \Phi^{01}\\ \Phi^{10} & \Phi^{11} \end{array}\right),$$
where Φ*^SS’^*_(*n*)_ = Φ*^S’S^*_(−*n*)_, ∑Φ^00^_(*n*)_ = 0, ∑Φ^10^_(*n*)_ – ∑Φ^01^_(*n*)_ = 0.

This model is well suited for describing tightly packed structures. At the same time, the continuous medium initiating the appearance of acoustic signals is not always homogeneous, which is the case, for example, for polymers and fibrous materials.

A visualisation of the considered models of initiation of acoustic signals of complex media under load is presented in Figure 21.

Oscillations of particles in a cell characterise the high-frequency component of the acoustic signal, while oscillations of the centre of mass characterise its low-frequency component. An increase in the size and structure of the cell allows us to move from a discrete to a continuous model of the medium. The results obtained make it possible to establish the boundaries of the use of discrete representations of changes in the structure of materials and the continuum model of the propagation of acoustic vibrations in a medium [40].

The use of acoustic measurement instruments makes it possible to find previously unknown characteristics of the structure of materials, to trace the dynamics of their development, stability, integrity, and to predict the development of material defects.

When evaluating the information content of diagnostic parameters, their sensitivity is determined by the following relation:(14)$$E=\frac{D_{2}-D_{1}}{D_{1}}/\frac{C_{2}-C_{1}}{C_{1}},$$
where *C* is the structural parameter, and *D* is the diagnostic parameter.

The more responsive diagnostic parameter *D* is to changes in the structural parameter, the higher its sensitivityeven more so at an early stage, making it possible to determine the malfunction.

Informatisation of the process of diagnostics of structures of materials is based on modern computer technologies that can quickly analyse information, with the issuance of recommendations for solving the tasks that contribute to a significant increase in the efficiency and reliability of the control of metal structures.

When informing the diagnostic process, not only the object itself is analysed, but also the reasons that caused deviations of the object’s properties from the established parameters, the mechanisms of the emergence and propagation of signals from defects, and ways to improve production technology and its control.

The functional purposes and components of the informatisation of the diagnostic process are presented in Figure 22 [41]. 

In the presented classification, there is an interaction of structure-forming elements, subordinated to the single goal of ensuring the operational reliability of structures.

Methods of informational diagnostics are an integral part of systems for identifying the defective structure of materials during their loading and operation, in which information is considered as a material object, which is characterised by a large amount of input data accumulated during structural changes.

## 5. Conclusions:

The method of non-contact deformation measurements allows us to take a fundamental step in determining the technical states of objects. The registration of vibration signals during deformation allows us to understand the mechanisms responsible for the accumulation of damage more clearly. The occurrence of vibration signals under loading can be used as an information parameter in the diagnosis of developing defects. 

It was established for the first time that fixing the moments of occurrence of vibrational signals and determining the number of excesses in the amplitudes of harmonics over the discrimination level allows structural models to be studied for diagnosing the strength properties of materials under dynamic loading of solids.

Transferring the moments of occurrence of vibrational signals to the strain diagram from the applied load allows us to trace the development of the physical processes of the hardening and destruction of materials.

The presented results of the theoretical and experimental study of the energy spectrum of vibration signals during deformation in the models of a continuous medium, the informational parameters of which are the operators of elastic energy, show that the violation of internal bonds between the translational and rotational properties of the continuous medium model in the form of a diatomic cell connected by elastic bonds initiates the oscillatory properties of the precursors of the destruction of materials of structures that are under load.

The proposed measurement method of vibration signals under uniaxial loading of materials using the coherent properties of laser radiation increases the information content of measurements and allows for the automation of the process of recording and processing information and for the monitoring of the continuous dynamic process of the loading of samples. 

## Figures and Tables

**Figure 1 materials-13-02046-f001:**
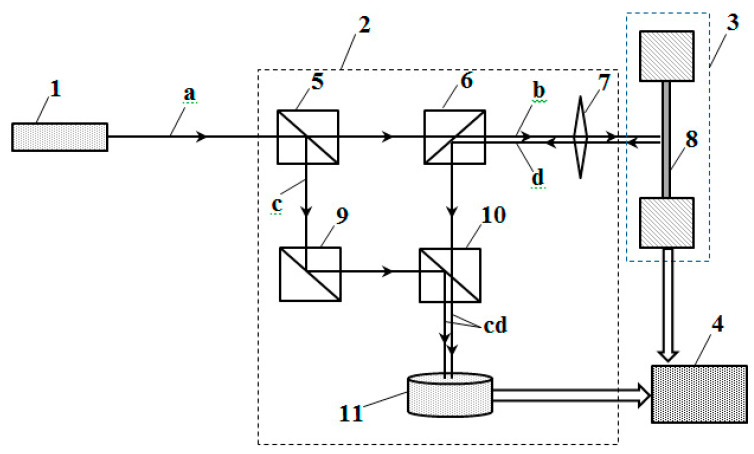
The structural diagram of a device for measuring the moments of occurrence of vibrational signals and their digital processing: 1—laser source, 2—interferometer, 3—loading device, 4—electronic unit for digital processing and data representation, 5, 6, 9, and 10—translucent mirrors, 7—lenses, 8—test sample, and 11—photodetector.

**Figure 2 materials-13-02046-f002:**
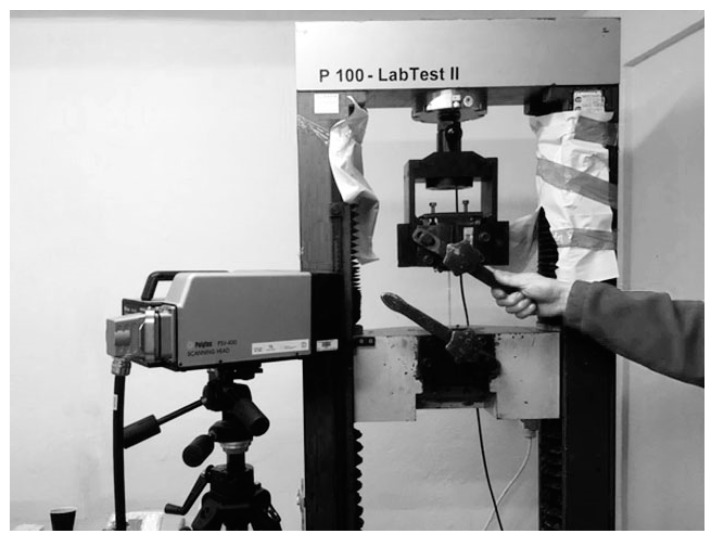
The general view of the experimental setup.

**Figure 3 materials-13-02046-f003:**
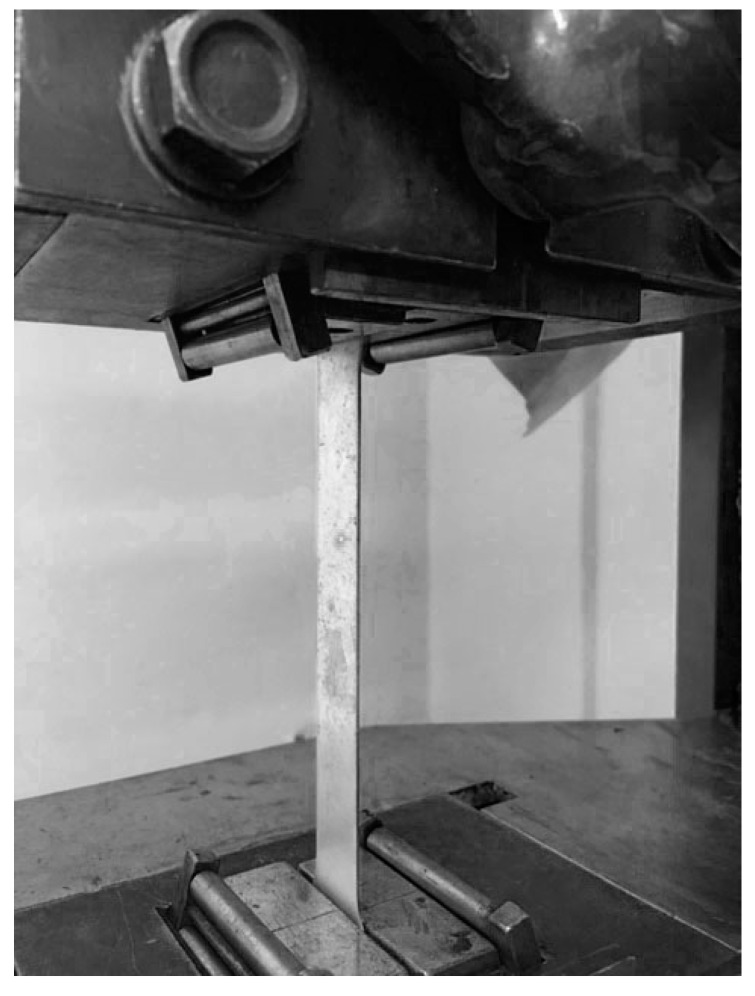
The location of the sample during the experiment.

**Figure 4 materials-13-02046-f004:**
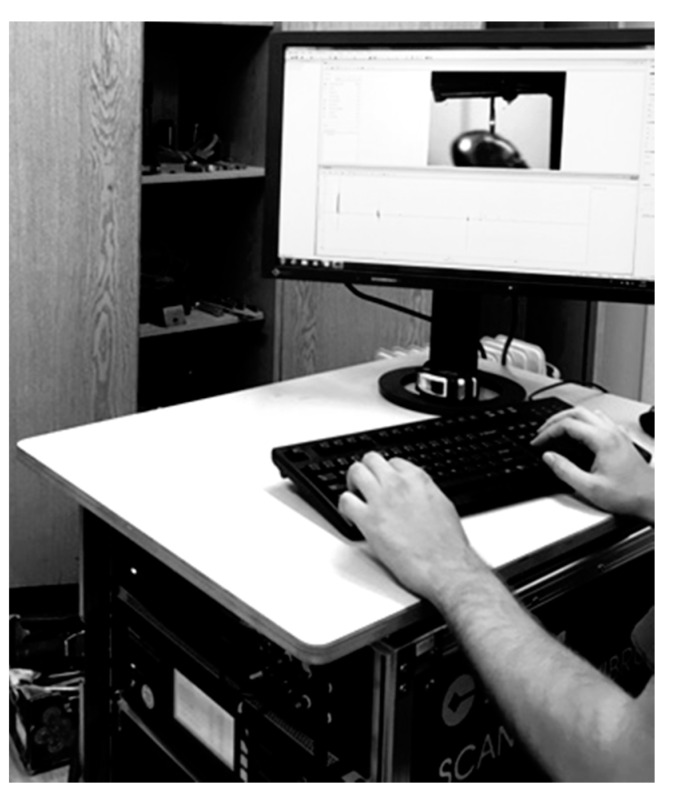
The fixation of the vibration signals.

**Figure 5 materials-13-02046-f005:**
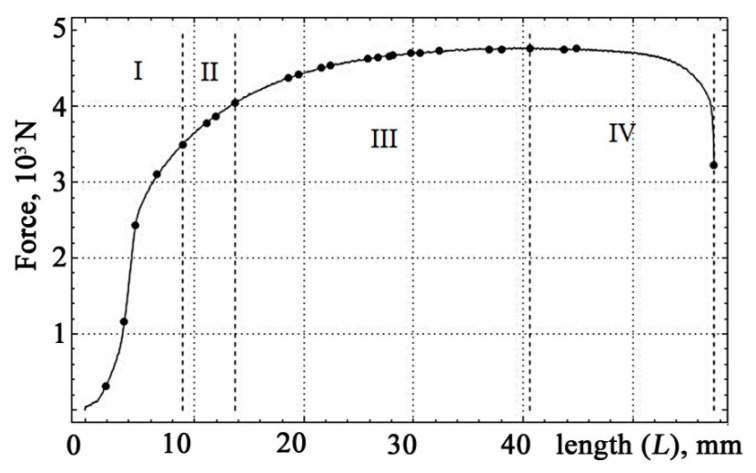
The occurrence moments of vibration signals.

**Figure 6 materials-13-02046-f006:**
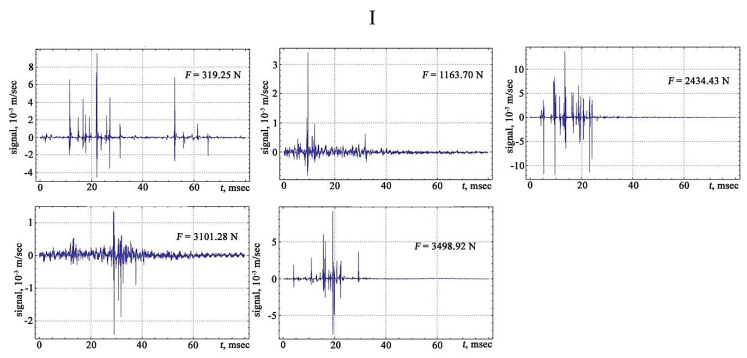
The vibration signals in different parts of the chart of stretching: I—plot of the elastic region.

**Figure 7 materials-13-02046-f007:**
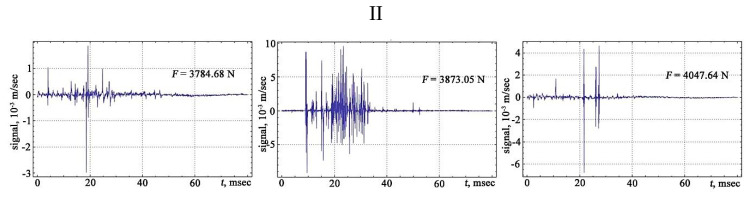
The vibration signals in different parts of the chart of stretching: II—area of the plastic flow of the material.

**Figure 8 materials-13-02046-f008:**
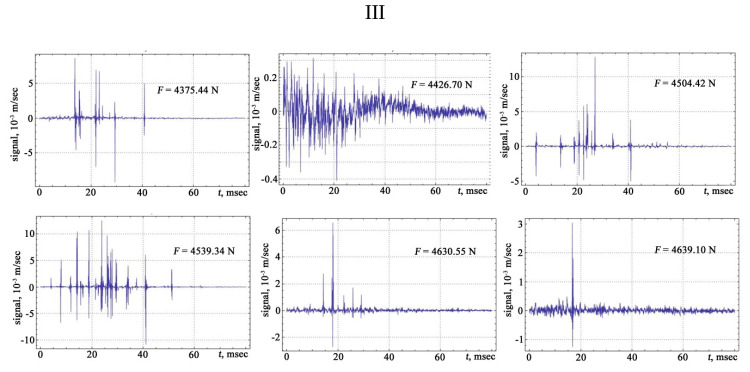
The vibration signals in different parts of the chart of stretching: III—phase of hardening

**Figure 9 materials-13-02046-f009:**
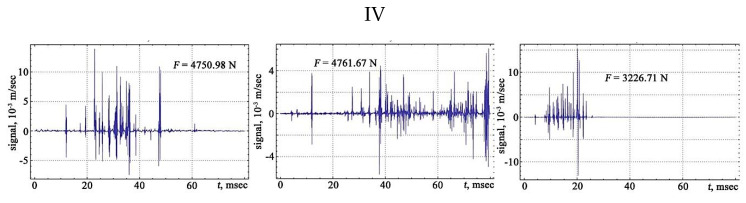
The vibration signals in different parts of the chart of stretching: IV—phase prior to the rupture of the sample.

**Figure 10 materials-13-02046-f010:**
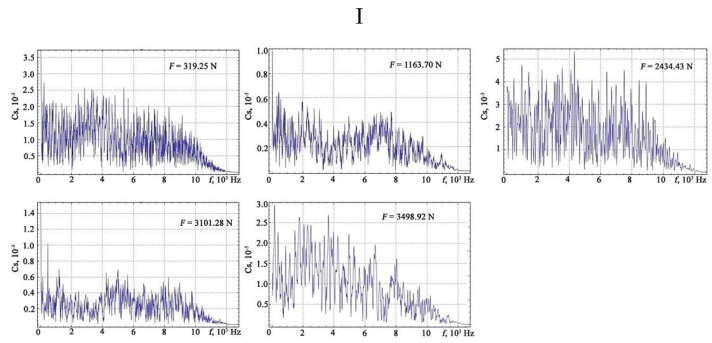
The results of the discrete Fourier transform of the vibration signals under different loads: I—plot of the elastic region.

**Figure 11 materials-13-02046-f011:**
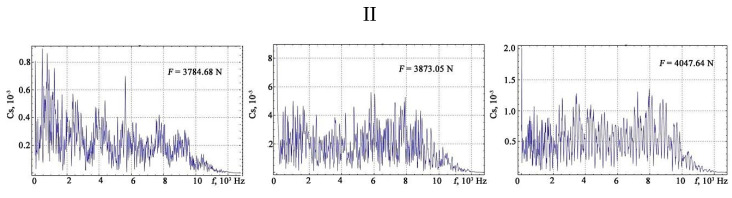
The results of the discrete Fourier transform of the vibration signals under different loads: II—area of the plastic flow of the material.

**Figure 12 materials-13-02046-f012:**
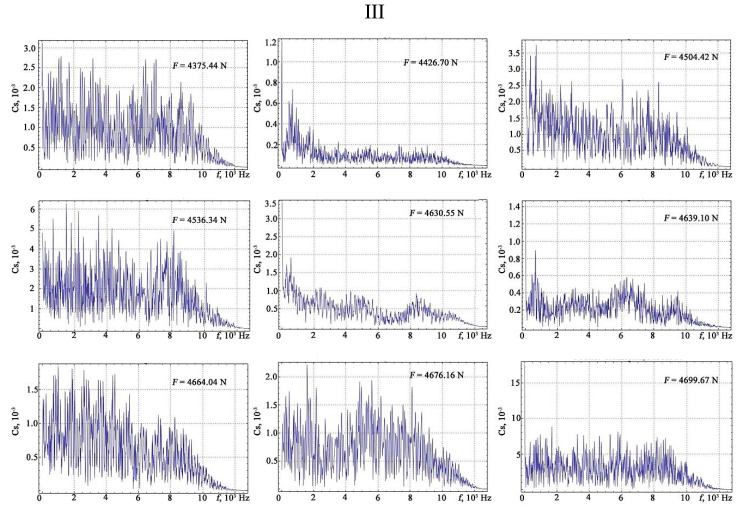
The results of the discrete Fourier transform of the vibration signals under different loads: III—phase of hardening.

**Figure 13 materials-13-02046-f013:**
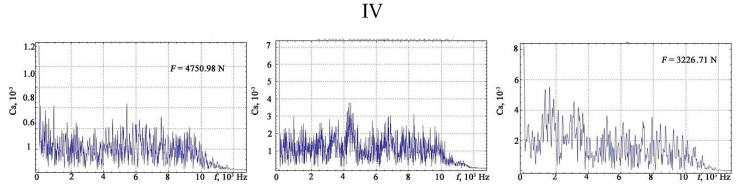
The results of the discrete Fourier transform of the vibration signals under different loads: IV—phase prior to the rupture of the sample.

**Figure 14 materials-13-02046-f014:**
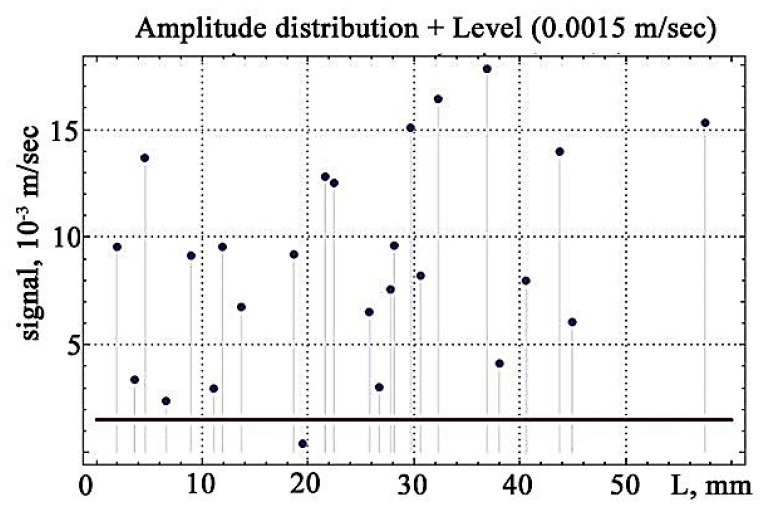
The distribution of the amplitudes of the vibration signals while increasing the applied load.

**Figure 15 materials-13-02046-f015:**
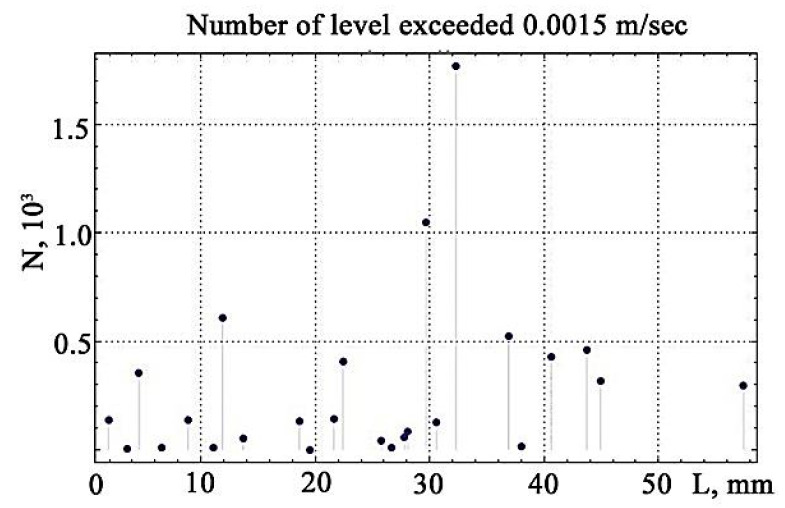
The total number of excess vibration amplitudes.

**Figure 16 materials-13-02046-f016:**
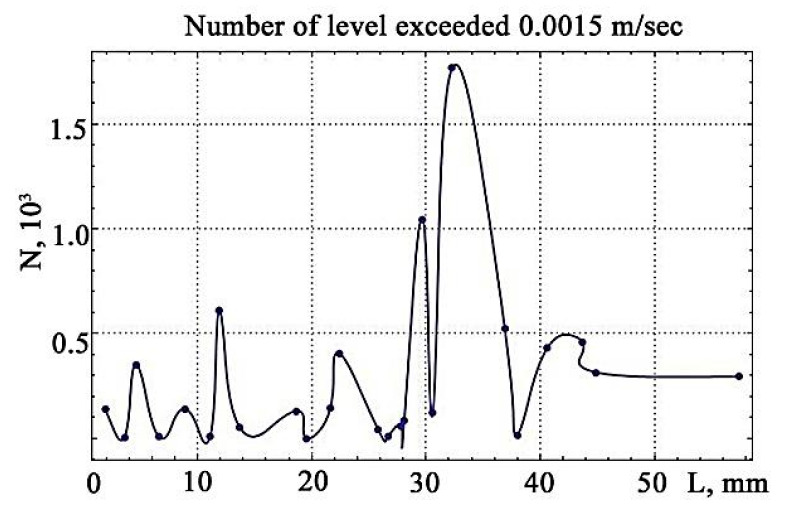
The visualisation of the number of excesses of vibration signals over the levels of discrimination.

**Figure 17 materials-13-02046-f017:**
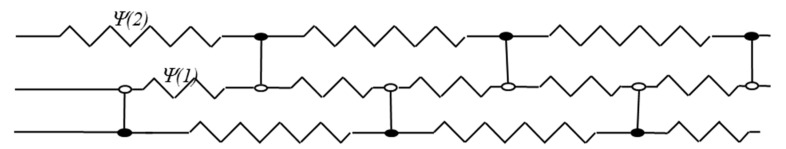
The one-dimensional model of discrete microstructure.

**Figure 18 materials-13-02046-f018:**
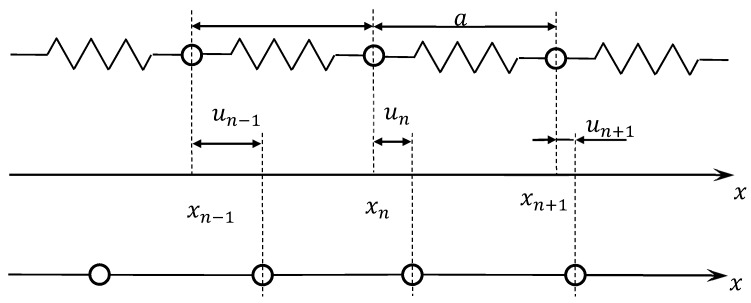
The propagation of a longitudinal disturbance in a chain of atoms.

**Figure 19 materials-13-02046-f019:**
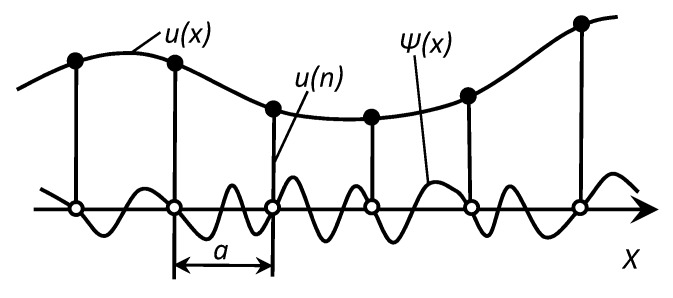
The one-dimensional quasicontinuum.

**Figure 20 materials-13-02046-f020:**
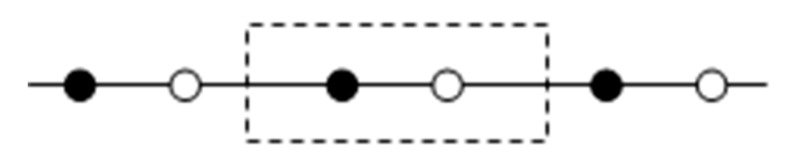
The discrete complex structure model.

**Figure 21 materials-13-02046-f021:**
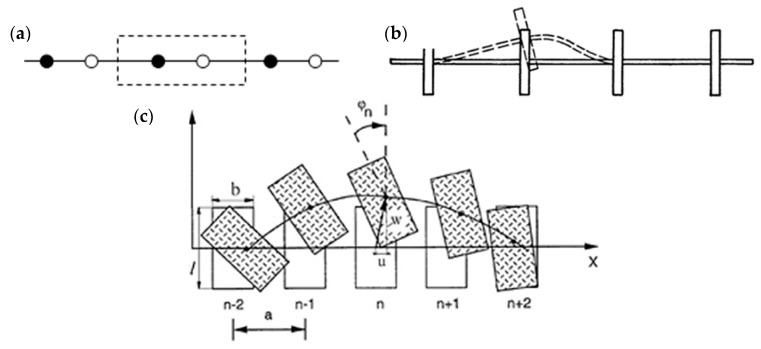
The models of the initiation of acoustic signals in complex environments: (**a**) model of a discrete complex structure, (**b**) Cosserat model, (**c**) model for initiating signals.

**Figure 22 materials-13-02046-f022:**
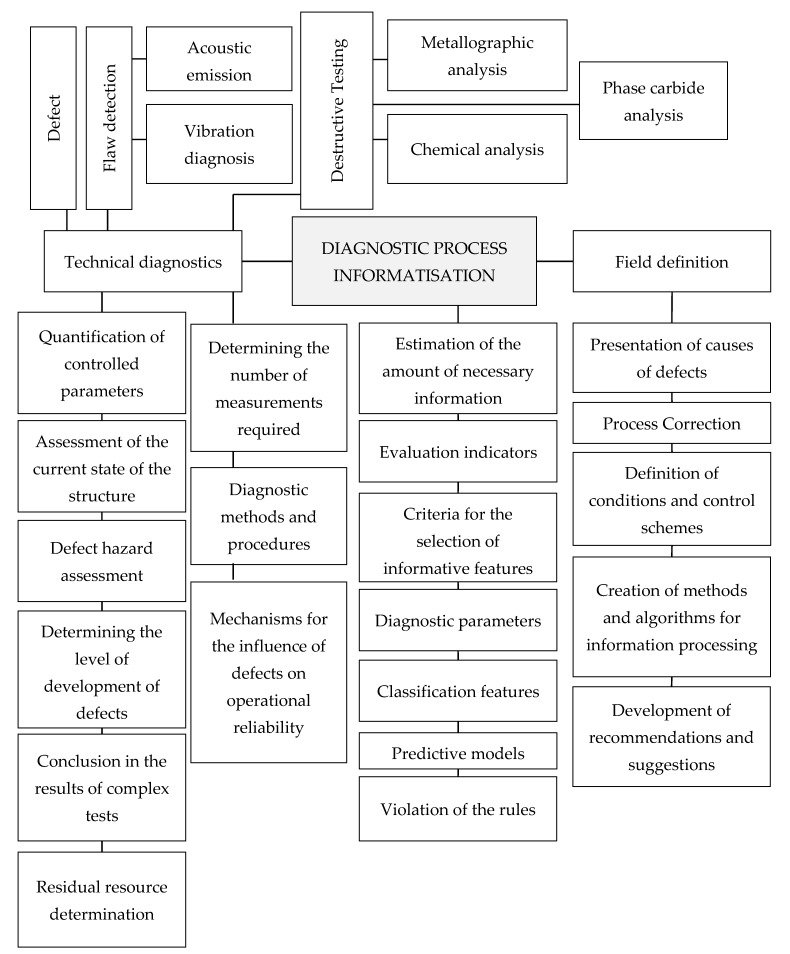
The functional purposes and components of the informatisation of the diagnostic process.

**Table 1 materials-13-02046-t001:** Quantitative values of absolute elongation corresponding to the occurrence moments of vibration signals for sheet steel, in mm.

Vibration Signals, m/s							
Occurrence of moment of vibration signals							
**Absolute elongation, mm**	1.7	1.8	1.9	**1.9**	4.3	6.1	5.7
	6.4	4.3	4.9	**3.6**	6.2	6.8	6.6
	12.6	8.5	6.4	**4.6**	7.4	7.7	8.1
	16.5	10.4	8.6	**6.6**	11.2	8.1	10.2
	20.6	13.5	9.2	**8.9**	13.7	8.8	16.3
	23.4	15.6	9.4	**11.1**	16.7	9.9	17.6
	26.0	18.5	11.8	**11.9**	17.8	10.7	19.9
	27.9	20.1	15.9	**13.7**	19.0	11.8	21.5
	29.6	21.1	19.2	**18.6**	22.4	14.7	21.8
	31.7	22.8	23.4	**19.5**	24.5	20.6	23.6
	33.0	24.0	24.4	**21.6**	26.7	22.0	24.4
	35.8	25.8	25.3	**22.4**	32.6	22.9	25.6
	37.4	27.3	26.4	**25.8**	35.5	27.6	26.4
	40.2	30.2	29.1	**26.7**	40.0	32.9	28.0
	41.5	32.1	34.4	**27.8**	41.4	38.0	29.3
	43.3	33.4	38.4	**28.1**	42.2	41.1	32.9
	45.5	34.2	43.7	**29.7**	gap	55.2	36.3
	47.8	36.9	48.3	**30.6**		gap	39.6
	49.0	38.9	gap	**32.3**			41.6
	52.8	42.6		**36.9**			43.2
	57.7	44.9		**38.0**			44.4
	60.5	46.6		**40.6**			49.8
	gap	48.4		**43.7**			52.2
		50.4		**44.9**			gap
		53.3		**61.0**			
		54.2		**gap**			
		55.1					
		56.0					
		gap					

**Table 2 materials-13-02046-t002:** Quantitative values of absolute elongation corresponding to the occurrence moments of vibration signals for dry wood and plastic.

	Moment Occurrence of the Vibration Signals	
	Dry Wood	Plastic
**Absolute elongation, mm**	1.2	2.3
	2.2	4.6
	6.4	5.0
	Gap	11.5
		Gap

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
