# Peer review of "Analysis of the Generation of Vibration Signals under Uniaxial Loading of Materials Using the Coherent Properties of Laser Radiation"

_materials, 2020, doi:10.3390/ma13092046_

Round 1

Reviewer 1 Report

The article is valuable due to the presentation and connection of the results of practical and theoretical research.

I have no comments and suggestions for Authors.

In my opinion, it can be published in the present form.

Author Response

Thanks to the reviewer for your support and attention.

Reviewer 2 Report

Manuscript (materials-743977) entitled “Analysis of Generation of Vibration Signals Under Uniaxial Loading of Materials with Using Coherent Properties of Laser Radiations” representing a research article was submitted to the MDPI Materials Journal.

This work contains twenty pages including sixteen figures, two tables, and 41 references. Fir their investigations, the authors used the method based on remote precision measurements of the amplitudes of harmonics of vibration signals and synchronous fixation of deformations under load. This work can be interesting for theoreticians and experimentalists as well as for graduate and postgraduate students.

The reviewer has looked through the paper and found that this paper requires a serious revision (major revision).

For instance, the following corrections of the English language can be done:

1) the first lines in Abstract: the first sentence does not contain a verb and it should be improved;

2) page 1, the last 4 lines: “allows not only detection of defects and establishment of the degree of

their danger but also evaluation of the stress-strain state” instead of “allows not only to detect defects and establish the degree of their danger, but also to evaluate the stress-strain state”;

3) page 2, the first lines: “Method of vibration diagnostics allows determination of the degree” instead of “Method of vibration diagnostics, allows to determine the degree”;

4) page 2, the last lines: “time but also” instead of “time, but also”;

5) page 3, in figure 1 title: “Figure 1. The structural diagram” instead of “Figure 1. Structural diagram”;

6) page 4, in figure 2 title: “Figure 2. The general view” instead of “Figure 2. General view”;

7) page 4, in figure 3 title: “Figure 3. The location” instead of “Figure 3. Location”;

8) page 5, in figure 4 title: “Figure 4. The fixation” instead of “Figure 4. Fixation”;

9) page 6, in figure 5 title: “Figure 5. The moments” instead of “Figure 5. Moments”;

10) line 3 after figure 7: “equal to 0.2 Csmax” instead of “equal 0,2Csmax”;

11) page 17, the last lines: “allows the increase” instead of “allows to increase”;

12) etc.

Table 1 must be on the same page. In Table 2, use smaller pt for the font size because some text is outside. Figure 6 is situated on two pages and figure 7 is also situated on two pages. It must exist a presentation culture and such large figures consisting of many figures must be divided.

Equations (3), (4), (8), (11), (12) contain symbol “&” over that must be deleted.

The English language must be polished. The corresponding figures and tables must be improved. The paper requires a major revision.

Author Response

Point 1: Corrected separate sections English text:

1) the first lines in Abstract: the first sentence does not contain a verb and it should be improved;

2) page 1, the last 4 lines: “allows not only detection of defects and establishment of the degree of

their danger but also evaluation of the stress-strain state” instead of “allows not only to detect defects and establish the degree of their danger, but also to evaluate the stress-strain state”;

3) page 2, the first lines: “Method of vibration diagnostics allows determination of the degree” instead of “Method of vibration diagnostics, allows to determine the degree”;

4) page 2, the last lines: “time but also” instead of “time, but also”;

5) page 3, in figure 1 title: “Figure 1. The structural diagram” instead of “Figure 1. Structural diagram”;

6) page 4, in figure 2 title: “Figure 2. The general view” instead of “Figure 2. General view”;

7) page 4, in figure 3 title: “Figure 3. The location” instead of “Figure 3. Location”;

8) page 5, in figure 4 title: “Figure 4. The fixation” instead of “Figure 4. Fixation”;

9) page 6, in figure 5 title: “Figure 5. The moments” instead of “Figure 5. Moments”;

10) line 3 after figure 7: “equal to 0.2 Csmax” instead of “equal 0,2Csmax”;

11) page 17, the last lines: “allows the increase” instead of “allows to increase”;

12) completed correction of the entire English text of the article

Point 2: Completed  adjustment of the structure of the article text, as a result of which table 1 is placed on one page, the pt of table 2 is changed. Figures 6 and 7 have additional figure captions.

Point 3: In equations (3), (4), (8), (11), (12) the symbol “&” is excluded

Point 4: The English language polished. The corresponding figures and tables improved.

Reviewer 3 Report

In this paper the method based on remote precision measurements of the amplitudes of harmonics of vibration signals and synchronous fixation of deformations under load is used. It was established that fixing the moments of occurrence of vibrational signals and determining the number of excesses in the amplitudes of harmonics over the discrimination level allows us to study structural models for diagnosing the strength properties of materials under dynamic loading of solids. Transferring the moments of occurrence of vibrational signals to the strain diagram from the applied load allows us to trace the development of physical processes of hardening and destruction of materials. The occurrence of vibrational signals under load can be used as an information parameter for the diagnosis of developing defects in the structure of materials under loading.

This paper is very interesting, research design is appropriate, the methods are adequately described and the results are clearly presented.

I think this paper is original work and after minor revision can be published in the present form.

I suggest that for the sake of clarity, Table 1 should be displayed on one page.

Equation (3), (4), (8), (11), (12) need to be corrected.

Author Response

Point 1: Completed  adjustment of the structure of the article text, as a result of which table 1 is placed on one page.

Point 2: In equations (3), (4), (8), (11), (12) the symbol “&” is excluded

Reviewer 4 Report

In this research, the authors put forward a method to investigate the force response f materials to external tensile stresses. Base on theory investigation, experimental data is used to verify the effectiveness of this method. However, there are following problems.

  1. The structure of the paper is not good. It is a little to understand.
  2. For the section introduction, the authors should give reference demonstration in detail, rather that [1-5], [16-22],[23-30].
  3. The format of the paper should be considered,
  4. The basic theory of this research is not clear. More demonstration should be given. It is better in section 2.
  5. The abstract should be rewritten.

Author Response

Point 1: Article structure improved

Point 2: Reference information added in detail.

“Much attention is paid to the diagnosis of damage and monitoring the development of defects In [6] a Kalman filter-based algorithm was used to measure the parameters of moving objects; in [7] diagnostic features were used for the condition monitoring of hypoid gear utilising the wavelet transform; in [8] location acoustic signals were used in a power transformer by fuzzy adaptive; in [9] improved least square generative adversarial networks were applied for rail crack detection by the ae technique; in [10] energy distribution and fractal characterisation of acoustic emission during coal deformation and fracturing were employed

Reviewed applicability of methods to solve specific technical problems. In [16] rolling element bearing defect frequencies were verified by vibration measurements on bearings with artificial faults on the outer/inner rings; in [17,18] gas turbine introduction and vibration diagnostic basics and soft-computing methods were employed; in [19,20,21]  a system of compressor units defects were diagnosed by the vibration value and a hydraulic rotator; and in [22]  vibration diagnostics of plates of knife refiners were performed

Are offered various technical improvements and original solutions In [23,24] optical coherence tomography with real-time profilometry of multiple inner layers and monitoring of steel shaft headgears were performed; in [25,26,27] symmetrised dot pattern and density-based clustering, frequency ratios to diagnose structural damage and auto-encoders were used; in [28,29,30] deep random forest fusion was used, polymer composite materials were developed, the full spectrum, and a dual channel was measured.”

Point 3: Document format revised

Point 4: Section 2. Materials and Methods added a paragraph explaining the essence of the method.

“Theoretical studies of the occurrence of acoustic signals with changes in the materials structure are developed in two main areas: the development of discrete models and models of the elastic continuum. Continuous models reflect free damped vibrations when scattered over local inhomogeneities and the propagantion boundaries of acoustic signals. The periodicity of the crystal lattice, its structure, and the constitution of atoms determine the features of the energy spectra. With a known force field, by solving the equations of motion, it is possible to obtain information on changes in the materials structure that cannot be directly measured.”

Point 5: Abstract rewritten and added text

“It was determined that shape of the propagating acoustic signal depends not only on the bias time, but also on the frequency.”

Round 2

Reviewer 2 Report

The authors have corrected only the half of recommended corrections.

For instance, in equations (3), (4), (8), (11), (12) the symbol “&” was not excluded.

Also, the authors should check the first line s after all equations and use verbs insted of "-".

It is necessary to use "The..." in every figure title after 'Figure (number of tye figure)."

The references should be in the MDPI format but not in a Chekh format.

The English language requires to be seriously polished . So, it is still a major revision.

Author Response

Point 1: In the previous version of the article in equations (3), (4), (8), (11), (12) we could not detect the "&" symbol and exclude it accordingly. The elimination of subscripts in these formulas did not result in the elimination of the "&"symbol. Since you write that this symbol is not missing we decided that this is a problem with the MathType 3 formula editor. It was decided to reprint the formulas in the newer MathType 6 editor. We hope this symbol disappears.

Point 2: We tried to correct the remaining comments as much as possible

("-" characters changed to verbs; was used "The..." in figure title; references has a MDPI format)

Point 3: English was also improved as much as possible

Reviewer 4 Report

Based on the modification, the paper has been much improved. It is much better compared with the orignal one.

  1. Figure 7 and Figure 8 are not clear. I suggest the authors modify it.
  2. It is not clear for Eq.(3), (4), and (8). It is better to modify it.
  3. It is better to improve the quality of the conclusions.

Author Response

Point 1: According to your comments, figures 6 and 7 are presented in a more understandable form. Namely they are divided into zones of material hardening

Point 2: Equations 3,4 and 8 are presented in a more modified form, where the summation indices are indicated.

Point 3: In accordance with your wishes, the conclusions are expanded. We hope that the quality has improved

Round 3

Reviewer 2 Report

The referee has reviewed the paper already the third time and found that many corrections recommended by the referee were not taking into account. For instance , equations (3), (4), (8), (11), and (12) still contain the sign "&".

Also, references are still inj a strange format. For instance, refrerebce [3] even contain the address of the MDPI Publisher, i.e. " Materials. 1. vyd. Basel: MDPI, St Alban-Anlage 66, Ch-4052 Basel, Switz"!!!

So, the paper still requires a major revision.

Author Response

Point 1: The "&" symbol has not yet been detected by us. We came to the logical conclusion that in your character library you cannot see superscript characters, namely ".". The superscript is an indicator of a partial derivative. We modified formulas (3), (4), (8), (11), (12). We hope the "&" symbol will finally disappear.

Point 2: Literature Corrected

Thank you very much for making and improving the quality and constructive comments.